# Robust Spectral Inference for Joint Stochastic Matrix Factorization

**Moontae Lee, David Bindel**
Dept. of Computer Science
Cornell University
Ithaca, NY 14850
{moontae,bindel}@cs.cornell.edu

**David Mimno**
Dept. of Information Science
Cornell University
Ithaca, NY 14850
mimno@cornell.edu

## Abstract

Spectral inference provides fast algorithms and provable optimality for latent topic analysis. But for real data these algorithms require additional ad-hoc heuristics, and even then often produce unusable results. We explain this poor performance by casting the problem of topic inference in the framework of Joint Stochastic Matrix Factorization (JSMF) and showing that previous methods violate the theoretical conditions necessary for a good solution to exist. We then propose a novel rectification method that learns high quality topics and their interactions even on small, noisy data. This method achieves results comparable to probabilistic techniques in several domains while maintaining scalability and provable optimality.

## 1 Introduction

Summarizing large data sets using pairwise co-occurrence frequencies is a powerful tool for data mining. Objects can often be better described by their relationships than their inherent characteristics. Communities can be discovered from friendships [1], song genres can be identified from co-occurrence in playlists [2], and neural word embeddings are factorizations of pairwise co-occurrence information [3, 4]. Recent *Anchor Word* algorithms [5, 6] perform spectral inference on co-occurrence statistics for inferring topic models [7, 8]. Co-occurrence statistics can be calculated using a single parallel pass through a training corpus. While these algorithms are fast, deterministic, and provably guaranteed, they are sensitive to observation noise and small samples, often producing effectively useless results on real documents that present no problems for probabilistic algorithms.

We cast this general problem of learning overlapping latent clusters as Joint-Stochastic Matrix Factorization (JSMF), a subset of non-negative matrix factorization that contains topic modeling as a special case. We explore the conditions necessary for inference from co-occurrence statistics and show that the Anchor Words algorithms necessarily violate such

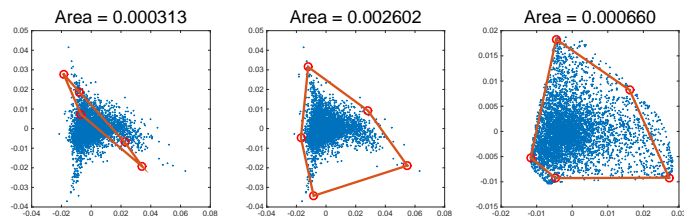

Figure 1: 2D visualizations show the low-quality convex hull found by Anchor Words [6] (left) and a better convex hull (middle) found by discovering anchor words on a rectified space (right).

conditions. Then we propose a rectified algorithm that matches the performance of probabilistic inference—even on small and noisy datasets—without losing efficiency and provable guarantees. Validating on both real and synthetic data, we demonstrate that our rectification not only produces better clusters, but also, unlike previous work, learns meaningful cluster interactions.

Let the matrix $C$ represent the co-occurrence of pairs drawn from $N$ objects: $C_{ij}$ is the joint probability $p(X_1 = i, X_2 = j)$ for a pair of objects $i$ and $j$. Our goal is to discover $K$ latent clusters by approximately decomposing $C \approx BAB^T$. $B$ is the object-cluster matrix, in which each column corresponds to a cluster and $B_{ik} = p(X = i | Z = k)$ is the probability of drawing an object $i$ conditioned on the object belonging to the cluster $k$; and $A$ is the cluster-cluster matrix, in which $A_{kl} = p(Z_1 = k, Z_2 = l)$ represents the joint probability of pairs of clusters. We call the matrices $C$ and $A$ *joint-stochastic* (i.e., $C \in \mathcal{JS}_N, A \in \mathcal{JS}_K$) due to their correspondence to joint distributions; $B$ is *column-stochastic*. Example applications are shown in Table 1.

Anchor Word algorithms [5, 6] solve JSMF problems using a separability assumption: each topic contains at least one "anchor" word that has non-negligible probability exclusively in that topic. The algorithm uses the co-occurrence

Table 1: JSMF applications, with anchor-word equivalents.

| Domain | Object | Cluster | Basis |
|---|---|---|---|
| Document | Word | Topic | Anchor Word |
| Image | Pixel | Segment | Pure Pixel |
| Network | User | Community | Representative |
| Legislature | Member | Party/Group | Partisan |
| Playlist | Song | Genre | Signature Song |

patterns of the anchor words as a summary basis for the co-occurrence patterns of all other words. The initial algorithm [5] is theoretically sound but unable to produce column-stochastic word-topic matrix $B$ due to unstable matrix inversions. A subsequent algorithm [6] fixes negative entries in $B$, but still produces large negative entries in the estimated topic-topic matrix $A$. As shown in Figure 3, the proposed algorithm infers valid topic-topic interactions.

## 2 Requirements for Factorization

In this section we review the probabilistic and statistical structures of JSMF and then define geometric structures of co-occurrence matrices required for successful factorization. $C \in \mathbb{R}^{N \times N}$ is a joint-stochastic matrix constructed from $M$ training examples, each of which contain some subset of $N$ objects. We wish to find $K \ll N$ latent clusters by factorizing $C$ into a column-stochastic matrix $B \in \mathbb{R}^{N \times K}$ and a joint-stochastic matrix $A \in \mathbb{R}^{K \times K}$, satisfying $C \approx BAB^T$.

**Probabilistic structure.** Figure 2 shows the event space of our model. The distribution $A$ over pairs of clusters is generated first from a stochastic process with a hyperparameter $\alpha$. If the $m$-th training example contains a total of $n_m$ objects, our model views the example as consisting of all possible $n_m(n_m - 1)$ pairs of objects.[1] For each of these pairs, cluster assignments are sampled from the selected distribution $((z_1, z_2) \sim A)$. Then an actual object pair is drawn with respect to the corresponding cluster assignments $(x_1 \sim B_{z_1}, x_2 \sim B_{z_2})$. Note that this process does not explain how each training example is generated from a model, but shows how our model understands the objects in the training examples.

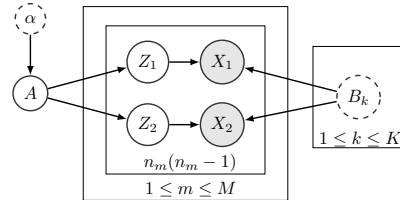

Figure 2: The JSMF event space differs from LDA's. JSMF deals only with pairwise co-occurrence events and does not generate observations/documents.

Following [5, 6], our model views $B$ as a set of parameters rather than random variables.[2] The primary learning task is to estimate $B$; we then estimate $A$ to recover the hyperparameter $\alpha$. Due to the conditional independence $X_1 \perp X_2 \mid (Z_1 \text{ or } Z_2)$, the factorization $C \approx BAB^T$ is equivalent to

$$p(X_1, X_2 | A; B) = \sum_{z_1} \sum_{z_2} p(X_1 | Z_1; B) p(Z_1, Z_2 | A) p(X_2 | Z_2; B).$$

Under the *separability assumption*, each cluster $k$ has a basis object $s_k$ such that $p(X = s_k | Z = k) > 0$ and $p(X = s_k | Z \neq k) = 0$. In matrix terms, we assume the submatrix of $B$ comprised of

the rows with indices $S = \{s_1, \ldots, s_K\}$ is diagonal. As these rows form a non-negative basis for the row space of $B$, the assumption implies $\text{rank}^+(B) = K = \text{rank}(B)$.[3] Providing identifiability to the factorization, this assumption becomes crucial for inference of both $B$ and $A$. Note that JSMF factorization is unique up to column permutation, meaning that no specific ordering exists among the discovered clusters, equivalent to probabilistic topic models (see the Appendix).

**Statistical structure.** Let $f(\alpha)$ be a (known) distribution of distributions from which a cluster distribution is sampled for each training example. Saying $W_m \sim f(\alpha)$, we have $M$ *i.i.d* samples $\{W_1, \ldots, W_M\}$ which are not directly observable. Defining the posterior cluster-cluster matrix $A_M^* = \frac{1}{M} \sum_{m=1}^{M} W_m W_m^T$ and the expectation $A^* = \mathbb{E}[W_m W_m^T]$, Lemma 2.2 in [5] showed that[4]

$$A_M^* \longrightarrow A^* \quad \text{as} \quad M \longrightarrow \infty. \tag{1}$$

Denote the *posterior co-occurrence* for the $m$-th training example by $C_m^*$ and all examples by $C^*$. Then $C_m^* = B W_m W_m^T B^T$, and $C^* = \frac{1}{M} \sum_{m=1}^{M} C_m^*$. Thus

$$C^* = B \left( \frac{1}{M} \sum_{m=1}^{M} W_m W_m^T \right) B^T = B A_M^* B^T. \tag{2}$$

Denote the *noisy observation* for the $m$-th training example by $C_m$, and all examples by $C$. Let $W = [W_1|...|W_M]$ be a matrix of topics. We will construct $C_m$ so that $\mathbb{E}[C|W]$ is an *unbiased estimator* of $C^*$. Thus as $M \to \infty$

$$C \longrightarrow \mathbb{E}[C] = C^* = B A_M^* B^T \longrightarrow B A^* B^T. \tag{3}$$

**Geometric structure.** Though the separability assumption allows us to identify $B$ even from the noisy observation $C$, we need to throughly investigate the structure of cluster interactions. This is because it will eventually be related to how much useful information the co-occurrence between corresponding anchor bases contains, enabling us to best use our training data. Say $\mathcal{DNN}_n$ is the set of $n \times n$ *doubly non-negative* matrices: entrywise non-negative and positive semidefinite (PSD).

**Claim** $A_M^*, A^* \in \mathcal{DNN}_K$ and $C^* \in \mathcal{DNN}_N$
**Proof** Take any vector $y \in \mathbb{R}^K$. As $A_M^*$ is defined as a sum of outer-products,

$$y^T A_M^* y = \frac{1}{M} \sum_{m=1}^{M} y^T W_m W_m^T y = \frac{1}{M} \sum (W_m^T y)^T (W_m^T y) = \sum (\textit{non-negative}) \geq 0. \tag{4}$$

Thus $A_M^* \in \mathcal{PSD}_K$. In addition, $(A_M^*)_{kl} = p(Z_1 = k, Z_2 = l) \geq 0$ for all $k, l$. Proving $A^* \in \mathcal{DNN}_K$ is analogous by the linearity of expectation. Relying on double non-negativity of $A_M^*$, Equation (3) implies not only the low-rank structure of $C^*$, but also double non-negativity of $C^*$ by a similar proof (see the Appendix).

The Anchor Word algorithms in [5, 6] consider neither double non-negativity of cluster interactions nor its implication on co-occurrence statistics. Indeed, the empirical co-occurrence matrices collected from limited data are generally indefinite and full-rank, whereas the posterior co-occurrences must be positive semidefinite and low-rank. Our new approach will efficiently enforce double non-negativity and low-rankness of the co-occurrence matrix $C$ based on the geometric property of its posterior behavior. We will later clarify how this process substantially improves the quality of the clusters and their interactions by eliminating noises and restoring missing information.

## 3 Rectified Anchor Words Algorithm

In this section, we describe how to estimate the co-occurrence matrix $C$ from the training data, and how to rectify $C$ so that it is low-rank and doubly non-negative. We then decompose the rectified $C'$ in a way that preserves the doubly non-negative structure in the cluster interaction matrix.

**Generating co-occurrence** $C$.  Let $H_m$ be the vector of object counts for the $m$-th training example, and let $p_m = BW_m$ where $W_m$ is the document's latent topic distribution. Then $H_m$ is assumed to be a sample from a multinomial distribution $H_m \sim \text{Multi}(n_m, p_m)$ where $n_m = \sum_{i=1}^{N} H_m^{(i)}$, and recall $\mathbb{E}[H_m] = n_m p_m = n_m BW_m$ and $\text{Cov}(H_m) = n_m \left( \text{diag}(p_m) - p_m p_m^T \right)$. As in [6], we generate the co-occurrence for the $m$-th example by

$$C_m = \frac{H_m H_m^T - \text{diag}(H_m)}{n_m (n_m - 1)}. \tag{5}$$

The diagonal penalty in Eq. 5 cancels out the diagonal matrix term in the variance-covariance matrix, making the estimator unbiased. Putting $d_m = n_m(n_m - 1)$, that is $\mathbb{E}[C_m | W_m] = \frac{1}{d_m} \mathbb{E}[H_m H_m^T] - \frac{1}{d_m} \text{diag}(\mathbb{E}[H_m]) = \frac{1}{d_m} (\mathbb{E}[H_m]\mathbb{E}[H_m]^T + \text{Cov}(H_m) - \text{diag}(\mathbb{E}[H_m])) = B(W_m W_m^T)B^T \equiv C_m^*$. Thus $\mathbb{E}[C|W] = C^*$ by the linearity of expectation.

**Rectifying co-occurrence** $C$.  While $C$ is an unbiased estimator for $C^*$ in our model, in reality the two matrices often differ due to a mismatch between our model assumptions and the data[5] or due to error in estimation from limited data. The computed $C$ is generally full-rank with many negative eigenvalues, causing a large approximation error. As the posterior co-occurrence $C^*$ must be low-rank, doubly non-negative, and joint-stochastic, we propose two rectification methods: Diagonal Completion (DC) and Alternating Projection (AP). DC modifies only diagonal entries so that $C$ becomes low-rank, non-negative, and joint-stochastic; while AP enforces modifies every entry and enforces the same properties as well as positive semi-definiteness. As our empirical results strongly favor alternating projection, we defer the details of diagonal completion to the Appendix.

Based on the desired property of the posterior co-occurrence $C^*$, we seek to project our estimator $C$ onto the set of joint-stochastic, doubly non-negative, low rank matrices. Alternating projection methods like Dykstra's algorithm [9] allow us to project onto an intersection of finitely many convex sets using projections onto each individual set in turn. In our setting, we consider the intersection of three sets of symmetric $N \times N$ matrices: the elementwise non-negative matrices $\mathcal{NN}_N$, the normalized matrices $\mathcal{NOR}_N$ whose entry sum is equal to 1, and the positive semi-definite matrices with rank $K$, $\mathcal{PSD}_{NK}$. We project onto these three sets as follows:

$$\Pi_{\mathcal{PSD}_{NK}}(C) = U\Lambda_K^+ U^T, \ \ \Pi_{\mathcal{NOR}_N}(C) = C + \frac{1 - \sum_{i,j} C_{ij}}{N^2} \mathbf{1}\mathbf{1}^T, \ \ \Pi_{\mathcal{NN}_N}(C) = \max\{C, \ 0\}.$$

where $C = U\Lambda U^T$ is an eigendecomposition and $\Lambda_K^+$ is the matrix $\Lambda$ modified so that all negative eigenvalues and any but the $K$ largest positive eigenvalues are set to zero. Truncated eigendecompositions can be computed efficiently, and the other projections are likewise efficient. While $\mathcal{NN}_N$ and $\mathcal{NOR}_N$ are convex, $\mathcal{PSD}_{NK}$ is not. However, [10] show that alternating projection with a non-convex set still works under certain conditions, guaranteeing a local convergence. Thus iterating three projections in turn until the convergence rectifies $C$ to be in the desired space. We will show how to satisfy such conditions and the convergence behavior in Section 5.

**Selecting basis** $S$.  The first step of the factorization is to select the subset $S$ of objects that satisfy the separability assumption. We want the $K$ best rows of the row-normalized co-occurrence matrix $\overline{C}$ so that all other rows lie nearly in the convex hull of the selected rows. [6] use the Gram-Schmidt process to select anchors, which computes *pivoted QR decomposition*, but did not utilize the sparsity of $\overline{C}$. To scale beyond small vocabularies, they use random projections that approximately preserve $\ell_2$ distances between rows of $\overline{C}$. For all experiments we use a new pivoted QR algorithm (see the Appendix) that exploits sparsity instead of using random projections, and thus preserves deterministic inference.[6]

**Recovering object-cluster** $B$.  After finding the set of basis objects $S$, we can infer each entry of $B$ by Bayes' rule as in [6]. Let $\{p(Z_1 = k | X_1 = i)\}_{k=1}^{K}$ be the coefficients that reconstruct the $i$-th row of $\overline{C}$ in terms of the basis rows corresponding to $S$. Since $B_{ik} = p(X_1 = i | Z_1 = k)$,

we can use the corpus frequencies $p(X_1 = i) = \sum_j C_{ij}$ to estimate $B_{ik} \propto p(Z_1 = k|X_1 = i)p(X_1 = i)$. Thus the main task for this step is to solve simplex-constrained QPs to infer a set of such coefficients for each object. We use an exponentiated gradient algorithm to solve the problem similar to [6]. Note that this step can be efficiently done in parallel for each object.

**Recovering cluster-cluster** $A$.
[6] recovered $A$ by minimizing $\|C - BAB^T\|_F$; but the inferred $A$ generally has many negative entries, failing to model the probabilistic interaction between topics. While we can further project $A$ onto the joint-stochastic matrices, this produces a large approximation error.

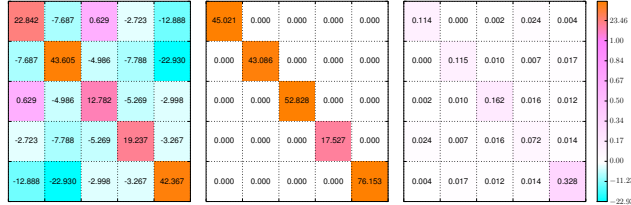

Figure 3: The algorithm of [6] (first panel) produces negative cluster co-occurrence probabilities. A probabilistic reconstruction alone (this paper & [5], second panel) removes negative entries but has no off-diagonals and does not sum to one. Trying after rectification (this paper, third panel) produces a valid joint stochastic matrix.

We consider an alternate recovery method that again leverages the separability assumption. Let $C_{SS}$ be the submatrix whose rows and columns correspond to the selected objects $S$, and let $D$ be the diagonal submatrix $B_{S*}$ of rows of $B$ corresponding to $S$. Then

$$C_{SS} = DAD^T = DAD \implies A = D^{-1}C_{SS}D^{-1}. \tag{6}$$

This approach efficiently recovers a cluster-cluster matrix $A$ mostly based on the co-occrrurence information between corresponding anchor basis, and produces no negative entries due to the stability of diagonal matrix inversion. Note that the principle submatrices of a PSD matrix are also PSD; hence, if $C \in \mathcal{PSD}_N$ then $C_{SS}, A \in \mathcal{PSD}_K$. Thus, not only is the recovered $A$ an unbiased estimator for $A_M^*$, but also it is now doubly non-negative as $A_M^* \in \mathcal{DNN}_K$ after the rectification.[7]

## 4  Experimental Results

Our Rectified Anchor Words algorithm with alternating projection fixes many problems in the baseline Anchor Words algorithm [6] while matching the performance of Gibbs sampling [11] and maintaining spectral inference's determinism and independence from corpus size. We evaluate direct measurement of matrix quality as well as indicators of topic utility. We use two text datasets: NIPS full papers and New York Times news articles.[8] We eliminate a minimal list of 347 English stop words and prune rare words based on tf-idf scores and remove documents with fewer than five tokens after vocabulary curation. We also prepare two non-textual item-selection datasets: users' movie reviews from the Movielens 10M Dataset,[9] and music playlists from the complete Yes.com dataset.[10] We perform similar vocabulary curation and document tailoring, with the exception of frequent stop-object elimination. Playlists often contain the same songs multiple times, but users are unlikely to review the same movies more than once, so we augment the movie dataset so that each review contains $2 \times (stars)$ number of movies based on the half-scaled rating information that varies from 0.5 stars to 5 stars. Statistics of our datasets are shown in Table 2.

Table 2: Statistics of four datasets.

| Dataset | $M$ | $N$ | Avg. Len |
|---|---|---|---|
| NIPS | 1,348 | 5k | 380.5 |
| NYTimes | 269,325 | 15k | 204.9 |
| Movies | 63,041 | 10k | 142.8 |
| Songs | 14,653 | 10k | 119.2 |

We run DC 30 times for each experiment, randomly permuting the order of objects and using the median results to minimize the effect of different orderings. We also run 150 iterations of AP alternating $\mathcal{PSD}_{NK}$, $\mathcal{NOR}_N$, and $\mathcal{NN}_N$ in turn. For probabilistic Gibbs sampling, we use the Mallet with the standard option doing 1,000 iterations. All metrics are evaluated against the original $C$, *not against the rectified $C'$*, whereas we use $B$ and $A$ inferred from the rectified $C'$.

[7]We later realized that essentially same approach was previously tried in [5], but it was not able to generate a valid topic-topic matrix as shown in the middle panel of Figure 3.

[8]https://archive.ics.uci.edu/ml/datasets/Bag+of+Words

[9]http://grouplens.org/datasets/movielens

[10]http://www.cs.cornell.edu/~shuochen/lme

**Qualitative results.** Although [6] report comparable results to probabilistic algorithms for LDA, the algorithm fails under many circumstances. The algorithm prefers rare and unusual anchor words that form a poor basis, so topic clusters consist of the same high-frequency terms repeatedly, as shown in the upper third of Table 3. In contrast, our algorithm with AP rectification successfully learns themes similar to the probabilistic algorithm. One can also verify that cluster interactions given in the third panel of Figure 3 explain how the five topics correlate with each other.

Similar to [12], we visualize the five anchor words in the co-occurrence space after 2D PCA of $\overline{C}$. Each panel in Figure 1 shows a 2D embedding of the NIPS vocabulary as blue dots and five selected anchor words in red. The first plot shows standard anchor words and the original co-occurrence space. The second plot shows anchor words selected from the rectified space overlaid on the original co-occurrence space. The third plot shows the same anchor words as the second plot overlaid on the AP-rectified space. The rectified anchor words provide better coverage on both spaces, explaining why we are able to achieve reasonable topics even with $K = 5$.

Table 3: Each line is a topic from NIPS ($K = 5$). Previous work simply repeats the most frequent words in the corpus five times.

| **Arora et al. 2013 (Baseline)** |
| --- |
| neuron layer hidden recognition signal cell noise |
| neuron layer hidden cell signal representation noise |
| neuron layer cell hidden signal noise dynamic |
| neuron layer cell hidden control signal noise |
| neuron layer hidden cell signal recognition noise |
| **This paper (AP)** |
| neuron circuit cell synaptic signal layer activity |
| control action dynamic optimal policy controller reinforcement |
| recognition layer hidden word speech image net |
| cell field visual direction image motion object orientation |
| gaussian noise hidden approximation matrix bound examples |
| **Probabilistic LDA (Gibbs)** |
| neuron cell visual signal response field activity |
| control action policy optimal reinforcement dynamic robot |
| recognition image object feature word speech features |
| hidden net layer dynamic neuron recurrent noise |
| gaussian approximation matrix bound component variables |

Rectification also produces better clusters in the non-textual movie dataset. Each cluster is notably more genre-coherent and year-coherent than the clusters from the original algorithm. When $K = 15$, for example, we verify a cluster of *Walt Disney 2D Animations* mostly from the 1990s and a cluster of Fantasy movies represented by *Lord of the Rings* films, similar to clusters found by probabilistic Gibbs sampling. The Baseline algorithm [6] repeats *Pulp Fiction* and *Silence of the Lambs* 15 times.

**Quantitative results.** We measure the intrinsic quality of inference and summarization with respect to the JSMF objectives as well as the extrinsic quality of resulting topics. Lines correspond to four methods: ○ Baseline for the algorithm in the previous work [6] without any rectification, △ DC for Diagonal Completion, □ AP for Alternating Projection, and ◇ Gibbs for Gibbs sampling.

Anchor objects should form a good basis for the remaining objects. We measure **Recovery** error $\left( \frac{1}{N} \sum_i^N \|\overline{C}_i - \sum_k^K p(Z_1 = k|X_1 = i)\overline{C}_{S_k}\|_2 \right)$ with respect to the original $C$ matrix, *not* the rectified matrix. AP reduces error in almost all cases and is more effective than DC. Although we expect error to decrease as we increase the number of clusters $K$, reducing recovery error for a fixed $K$ by choosing better anchors is extremely difficult: no other subset selection algorithm [13] decreased error by more than 0.001. A good matrix factorization should have small element-wise **Approximation** error $\left( \|C - BAB^T\|_F \right)$. DC and AP preserve more of the information in the original matrix $C$ than the Baseline method, especially when $K$ is small.[11] We expect non-trivial interactions between clusters, even when we do not explicitly model them as in [14]. Greater diagonal **Dominancy** $\left( \frac{1}{K} \sum_k^K p(Z_2 = k|Z_1 = k) \right)$ indicates lower correlation between clusters.[12] AP and Gibbs results are similar. We do not report held-out probability because we find that relative results are determined by user-defined smoothing parameters [12, 24].

**Specificity** $\left( \frac{1}{K} \sum_k^K KL\left(p(X|Z = k)\|p(X)\right) \right)$ measures how much each cluster is distinct from the corpus distribution. When anchors produce a poor basis, the conditional distribution of clus-

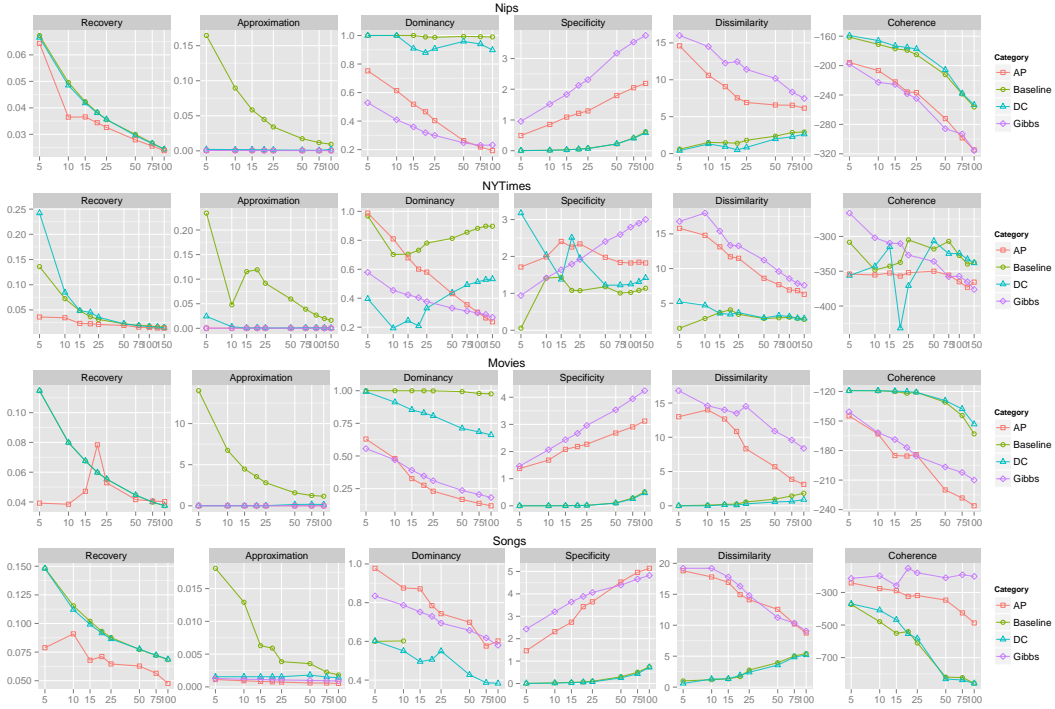

Figure 4: Experimental results on real dataset. The x-axis indicates $logK$ where $K$ varies by 5 up to 25 topics and by 25 up to 100 or 150 topics. Whereas the Baseline algorithm largely fails with small $K$ and does not infer quality $B$ and $A$ even with large $K$, Alternating Projection (AP) not only finds better basis vectors (Recovery), but also shows stable and comparable behaviors to probabilistic inference (Gibbs) in every metric.

ters given objects becomes uniform, making $p(X|Z)$ similar to $p(X)$. Inter-topic **Dissimilarity** counts the average number of objects in each cluster that do not occur in any other cluster's top 20 objects. Our experiments validate that AP and Gibbs yield comparably specific and distinct topics, while Baseline and DC simply repeat the corpus distribution as in Table 3. **Coherence** $\left(\frac{1}{K}\sum_k^K\sum_{x_1 \neq x_2}^{\in Top_k} \log \frac{D_2(x_1,x_2)+\epsilon}{D_1(x_2)}\right)$ penalizes topics that assign high probability (rank $> 20$) to words that do not occur together frequently. AP produces results close to Gibbs sampling, and far from the Baseline and DC. While this metric correlates with human evaluation of clusters [15] "worse" coherence can actually be better because the metric does not penalize repetition [12].

In **semi-synthetic experiments** [6] AP matches Gibbs sampling and outperforms the Baseline, but the discrepancies in topic quality metrics are smaller than in the real experiments (see Appendix). We speculate that semi-synthetic data is more "well-behaved" than real data, explaining why issues were not recognized previously.

## 5 Analysis of Algorithm

**Why does AP work?** Before rectification, diagonals of the empirical $C$ matrix may be far from correct. Bursty objects yield diagonal entries that are too large; extremely rare objects that occur at most once per document yield zero diagonals. Rare objects are problematic in general: the corresponding rows in the $C$ matrix are sparse and noisy, and these rows are likely to be selected by the pivoted QR. Because rare objects are likely to be anchors, the matrix $C_{SS}$ is likely to be highly diagonally dominant, and provides an uninformative picture of topic correlations. These problems are exacerbated when $K$ is small relative to the effective rank of $C$, so that an early choice of a poor anchor precludes a better choice later on; and when the number of documents $M$ is small, in which case the empirical $C$ is relatively sparse and is strongly affected by noise. To mitigate this issue, [24] run exhaustive grid search to find document frequency cutoffs to get informative anchors. As

model performance is inconsistent for different cutoffs and search requires cross-validation for each case, it is nearly impossible to find good heuristics for each dataset and number of topics.

Fortunately, a low-rank PSD matrix cannot have too many diagonally-dominant rows, since this violates the low rank property. Nor can it have diagonal entries that are small relative to off-diagonals, since this violates positive semi-definiteness. Because the anchor word assumption implies that non-negative rank and ordinary rank are the same, the AP algorithm ideally does not remove the information we wish to learn; rather, 1) the low-rank projection in AP suppresses the influence of small numbers of noisy rows associated with rare words which may not be well correlated with the others, and 2) the PSD projection in AP recovers missing information in diagonals. (As illustrated in the Dominancy panel of the Songs corpus in Figure 4, AP shows valid dominancies even after $K > 10$ in contrast to the Baseline algorithm.)

**Why does AP converge?** AP enjoys local linear convergence [10] if 1) the initial $C$ is near the convergence point $C'$, 2) $\mathcal{PSD}_{NK}$ is *super-regular* at $C'$, and 3) *strong regularity* holds at $C'$. For the first condition, recall that we rectified $C'$ by pushing $C$ toward $C^*$, which is the ideal convergence point inside the intersection. Since $C \to C^*$ as shown in (5), $C$ is close to $C'$ as desired. The prox-regular sets[13] are subsets of super-regular sets, so prox-regularity of $\mathcal{PSD}_{NK}$ at $C'$ is sufficient for the second condition. For permutation invariant $\mathcal{M} \subset \mathbb{R}^N$, the spectral set of symmetric matrices is defined as $\lambda^{-1}(\mathcal{M}) = \{X \in \mathcal{S}_N : (\lambda_1(X), \dots, \lambda_N(X)) \in \mathcal{M}\}$, and $\lambda^{-1}(\mathcal{M})$ is prox-regular if and only if $\mathcal{M}$ is prox-regular [16, Th. 2.4]. Let $\mathcal{M}$ be $\{x \in \mathbb{R}_n^+ : |supp(x)| = K\}$. Since each element in $\mathcal{M}$ has exactly $K$ positive components and all others are zero, $\lambda^{-1}(\mathcal{M}) = \mathcal{PSD}_{NK}$. By the definition of $\mathcal{M}$ and $K < N$, $P_{\mathcal{M}}$ is locally unique almost everywhere, satisfying the second condition almost surely. (As the intersection of the convex set $\mathcal{PSD}_N$ and the smooth manifold of rank $K$ matrices, $\mathcal{PSD}_{NK}$ is a smooth manifold almost everywhere.)

Checking the third condition a priori is challenging, but we expect noise in the empirical $C$ to prevent an irregular solution, following the argument of Numerical Example 9 in [10]. We expect AP to converge locally linearly and we can verify local convergence of AP in practice. Empirically, the ratio of average distances between two iterations are always $\leq 0.9794$ on the NYTimes dataset (see the Appendix), and other datasets were similar. Note again that our rectified $C'$ is a result of pushing the empirical $C$ toward the ideal $C^*$. Because approximation factors of [6] are all computed based on how far $C$ and its co-occurrence shape could be distant from $C^*$'s, all provable guarantees of [6] hold better with our rectified $C'$.

## 6 Related and Future Work

JSMF is a specific structure-preserving Non-negative Matrix Factorization (NMF) performing spectral inference. [17, 18] exploit a similar separable structure for NMF problmes. To tackle hyperspectral unmixing problems, [19, 20] assume *pure pixels*, a separability-equivalent in computer vision. In more general NMF without such structures, RESCAL [21] studies tensorial extension of similar factorization and SymNMF [22] infers $BB^T$ rather than $BAB^T$. For topic modeling, [23] performs spectral inference on third moment tensor assuming topics are uncorrelated.

As the core of our algorithm is to rectify the input co-occurrence matrix, it can be combined with several recent developments. [24] proposes two regularization methods for recovering better $B$. [12] nonlinearly projects co-occurrence to low-dimensional space via $t$-SNE and achieves better anchors by finding the exact anchors in that space. [25] performs multiple random projections to low-dimensional spaces and recovers approximate anchors efficiently by divide-and-conquer strategy. In addition, our work also opens several promising research directions. How exactly do anchors found in the rectified $C'$ form better bases than ones found in the original space $C$? Since now the topic-topic matrix $A$ is again doubly non-negative and joint-stochastic, can we learn super-topics in a multi-layered hierarchical model by recursively applying JSMF to topic-topic co-occurrence $A$?

## Acknowledgments

This research is supported by NSF grant HCC:Large-0910664. We thank Adrian Lewis for valuable discussions on AP convergence.

## Footnotes

[1]Due to the bag-of-words assumption, every object can pair with any other object in that example, except itself. One implication of our work is better understanding the self-co-occurrences, the diagonal entries in the co-occurrence matrix.

[2]In LDA, each column of B is generated from a known distribution $B_k \sim Dir(\beta)$.

[3] $\text{rank}^+(B)$ means the non-negative rank of the matrix B, whereas $\text{rank}(B)$ means the usual rank.

[4] This convergence is not trivial while $\frac{1}{M} \sum_{m=1}^{M} W_m \to \mathbb{E}[W_m]$ as $M \to \infty$ by the Central Limit Theorem.

[5]There is no reason to expect real data to be generated from topics, much less exactly $K$ latent topics.

[6]To effectively use random projections, it is necessary to either find proper dimensions based on multiple trials or perform low-dimensional random projection multiple times [25] and merge the resulting anchors.

[11]In the NYTimes corpus, $10^{-2}$ is a large error: each element is around $10^{-9}$ due to the number of normalized entries.

[12]Dominancy in Songs corpus lacks any Baseline results at $K > 10$ because dominancy is undefined if an algorithm picks a song that occurs at most once in each playlist as a basis object. In this case, the original construction of $C_{SS}$, and hence of $A$, has a zero diagonal element, making dominancy NaN.

[13]A set $\mathcal{M}$ is prox-regular if $P_{\mathcal{M}}$ is locally unique.

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
