[Supplementary Material]

# Robust Spectral Inference for Joint Stochastic Matrix Factorization (Supplementary Material)

**Moontae Lee, David Bindel**
Dept. of Computer Science
Cornell University
Ithaca, NY 14850
{moontae,bindel}@cs.cornell.edu

**David Mimno**
Dept. of Information Science
Cornell University
Ithaca, NY 14850
mimno@cornell.edu

## 1   Introduction

This is a supplementary document for the paper: Robust Spectral Inference for Joint Stochastic matrix Factorization. It is organized accordingly to the main paper so that the readers can find the missing proofs, deferred details, and further explanations in the corresponding sections. We also include more algorithms, experiments, and analysis that are discarded from the main paper due to the page limit.

## 2   Requirements for Factorization

**Proof for uniqueness of JSMF.**   When factorizing co-occurrence matrix $C$ into $BAB^T$ with constraints $B$: $N \times K$ column-stochastic and $A$: $K \times K$ joint-stochastic, the resulting $(B, A)$ may not be an unique decomposition of $C$ if $K \geq 2$. Assume there exists a $K \times K$ column-stochastic square matrix $Y$ such that $Y$ and $Y^{-1}$ are both non-negative. Then,

$$C \approx BAB^T = B(YY^{-1})A(YY^{-1})B^T = (BY)(Y^{-1}AY^{-T})(BY)^T. \tag{1}$$

As $BY$ is $N \times K$ column-stochastic and $Y^{-1}AY^{-T}$ is $K \times K$ joint-stochastic, $(BY, Y^{-1}AY^{-T})$ can be another equally meaningful solution for JSMF of $C$. In fact, it is known that if an inverse of non-negative matrix $Y$ is again non-negative, $Y$ must be a *generalized permutation matrix* which satisfies $Y = DP$ for some diagonal matrix $D$ and permutation matrix $P$. Since both $BY$ and $B$ are column-stochastic, $Y$ must be column-stochastic as well. Thus the only diagonal matrix $D$ that makes $DP$ column-stochastic with respect to permutation matrix $P$ is the identity matrix. Therefore we can conclude that the only possible $Y$ is a permutation matrix. It means that our factorization is **unique up to the column permutation**. This is equivalent to the fact that there is no order between resulting topics in probabilistic topic models.

**Proof for double non-negativity of posterior co-occurrence.**   Take any vector $y \in \mathbb{R}^N$ and say $y' = B^T y$. Then

$$y^T \mathbb{E}[C^*]y = y^T B A_M^* B^T y = (y')^T A_M^* y' \geq 0 \quad (\because A_M^* \in \mathcal{PSD}_K). \tag{2}$$

Thus $C^* \in \mathcal{PSD}_N$. Also, $C_{ij}^* = p(X_1 = i, X_2 = j) \geq 0$ for all $i, j$. Therefore $C^* \in \mathcal{DNN}_N$.

## 3   Rectified Anchor Word Algorithm

**Rectifying co-occurrence $C$ by Diagonal Completion (DC).**   As we explained in Section 6 of the main paper, the diagonal entries of the co-occurrence matrix are the most difficult elements to interpret and the least likely to conform to the model. For instance, frequent words in a document are

---

**Algorithm 1** Diagonal Completion (DC)

---

**In:** $F : (N/2) \times (N/2)$ block of $C$ in diagonal side
    $G : (N/2) \times (N/2)$ block of $C$ in off-diagonal side
**Out:** $d \in \mathbb{R}^{N/2}$ : a vector of new diagonal entries
**def** DIAGONAL-COMPLETE$(F, G)$
  $(U, \Sigma, V) \leftarrow truncated\text{-}svd(G, K)$
  $L \leftarrow U^T \times (F - F_{diag})$
  **for** $j = 1$ to $N/2$ **do**
    $u_j^T \leftarrow U_{j*}$
    $d_j \leftarrow \frac{1}{1 - \|u_j\|_2^2}(u_j^T \times L_{*j})$
  **end for**
  **return** $d$

---

likely to be bursty, leading to large diagonal elements; but popular songs appear at most a few times in any given playlist, leading to relatively small and noisy diagonal elements. Instead of ignoring such high variance, we fix the diagonal so that $C$ has low rank.

Algorithm 1 estimates the diagonal elements of $C$ from the off-diagonal elements, assuming the off-diagonal elements come from a low-rank matrix. The key observation is that the top or bottom halves of $C$ are themselves low rank, and we can find the range space for each matrix from those columns that are completely known. Once we know a space in which all the columns of the top half of $C$ should belong, we can determine the unknown diagonal elements through a least-squares fit using the known elements.

More concretely, the algorithm proceeds by partitioning $C$ into four quadrants of near-equal size. Let $F = C_{11}$ and $G = C_{12}$, and for each $j \in [1, N/2]$, let $J$ be all indices from 1 to $N/2$ except $j$. We want each column $F_{*j}$ to dwell in the range space of $G$. We find a basis $U$ for this range space from the first $K$ left singular vectors of $G$. To find $F_{jj}$, we seek a $K$-dimensional vector $y$ such that $(U_{J*})y = F_{Jj}$. Because we are unlikely to exactly satisfy this equation, we seek the least-square solution

$$(U_{J*})^T(U_{J*})y = (U_{J*})^T F_{Jj}. \tag{3}$$

Denote the $K \times K$ identity matrix by $I_K$ and $j$-th row vector of $U$ by $u_j^T$. Since $U$ has orthonormal columns, $(U_{J*})^T(U_{J*}) = I_K - u_j u_j^T$. By the Sherman-Morrison formula,

$$\left(I_K - u_j u_j^T\right)^{-1} = I_K + u_j u_j^T / (1 - u_j^T u_j). \tag{4}$$

Let $L_{*j} = (U_{J*})^T F_{Jj}$. Under the low rank assumption, the diagonal should be $d_j = (Uy)_j = (u_j^T)y$, and therefore

$$d_j = (u_j^T)\left(L_{*j} + \frac{u_j u_j^T}{1 - u_j^T u_j}L_{*j}\right) = \frac{u_j^T L_{*j}}{1 - u_j^T u_j}. \tag{5}$$

As we precompute $L$ and run the truncated SVD on a half-size block with $K \ll N$, DC is efficient. Simply execute Algorithm 1 twice with the inputs $(C_{11}, C_{12})$ and $(C_{22}, C_{21})$, and replace the existing diagonal with the output vector $e$. We present an error analysis in Section 6.

**Selecting basis S.**    After rectification, the next step is to select the subset $S$ of objects that satisfy the separability assumption. Our goal is to choose the $K$ best rows of the row-normalized co-occurrence matrix $\overline{C}$ so that all other rows lie nearly in the convex hull of the selected rows. [1] use the Gram-Schmidt process to select these anchor rows, but they do not use the sparsity of $\overline{C}$. In order to scale beyond relatively small vocabularies, they resort to random projections that approximately preserve $\ell_2$ distances.

Denote the row-normalized $C$ matrix by $\overline{C}$. Then by the conditional independence,

$$\overline{C}_{ij} = p(X_2 = j | X_1 = i) = \sum_{k'} p(X_2 = j | Z_1 = k')p(Z_1 = k' | X_1 = i). \tag{6}$$

**Algorithm 2** Finding Bases S

---

**In:** $P : N \times N$ matrix (e.g., $P \leftarrow \overline{C}^T$)
**Out:** $S$: the set of $K$ indices
   $r \in \mathbb{R}^K$: a vector of distances to each subspace
**def** FIND-S($P$)
   Initialize $S \leftarrow \varnothing$, $\;Q \leftarrow 0^{N \times K}$, $\;r \leftarrow 0^K$
   $norm \leftarrow$ squared norms of column vectors of $P$
   **for** $k = 1$ to $K$ **do**
      $n \leftarrow \text{argmax}_{1 \leq i \leq N} \, norm(i)$
      $S \leftarrow S \cup \{n\}$, $\;Q_{*k} \leftarrow P_{*n}$, $\;r_k \leftarrow \sqrt{norm(n)}$
      $Q_{*k} \leftarrow (Q_{*k} - \sum_{l=1}^{k-1} \langle Q_{*l}, P_{*n} \rangle Q_{*l}) / r_k$
      $norm \leftarrow norm - (Q_{*k}^T P) \circ (Q_{*k}^T P)$
   **end for**
   **return** $(S, r)$

---

($\circ$ operation is the Hadamard Product, a simple element-wise multiplication between two vectors)

Let $S = \{s_1, ..., s_K\}$ be the set of $K$ basis objects. Then $\overline{C}_{s_k, j} = p(X_2 = j | Z_1 = k)$ because the separability assumption implies

$$p(Z_1 = k' | X_1 = s_k) = \begin{cases} 1 & (\text{if } k' = k) \\ 0 & (\text{if } k' \neq k) \end{cases}. \tag{7}$$

Thus $\overline{C}_{ij} = \sum_k p(Z_1 = k | X_1 = i) \overline{C}_{s_k, j}$, which means every row vector of $\overline{C}$ can be represented by a convex combination of the row vectors corresponding to the basis objects.

The (unprojected) Gram-Schmidt process in [1] computes a *pivoted QR decomposition* [2]. Several other algorithms compute the same decomposition and exploit sparsity [3]. In particular, one can find the set $S$ with $O(NK)$ auxiliary space and $O(\text{nnz}(C)K)$ time without modifying $\overline{C}$; this has the advantage that $\overline{C}$ is unchanged in memory and ready for use in the recovery step. Algorithm 2 requires only $O(NK)$ space to store $Q$ doing every update implicitly rather than changing the original input matrix. Not modifying $\overline{C}$ in place has the additional advantage of leaving $\overline{C}$ unchanged in memory and ready for use in the recovery step. Note that we only return the set of indices $S$ corresponding to the basis objects and the diagonal entries $r$ of $R$ as their absolute values.

In Algorithm 2, $norm$ is a $N$-dimensional row vector that provides a criterion to greedily choose the next best column for column-pivoting. It is updated once at the end of each iteration because for each $1 \leq j \leq N$,

$$\|P_{*j} - \langle Q_{*k}, P_{*j} \rangle\|_2^2 = \langle P_{*j}, P_{*j} \rangle - \langle Q_{*k}, P_{*j} \rangle^2. \tag{8}$$

($norm$ was initialized as the first inner-product term). However, this greedy strategy is only one way of approaching the general problem of *subset selection*. Recent work on this subject includes [4, 5]. [6] present a CUR decomposition whose matrix factors consist of columns and rows of the input matrix. These alternate subset selection strategies were not designed for non-negative approximation; unlike ordinary pivoted QR, they will not necessarily recover the desired basis in the absence of noise. In the presence of noise and model error, however, these alternate selection strategies may merit further attention.

**Recovering cluster-example $W$.** Recall that the standard topic modeling consists of two inferences: inferring topic distributions in terms of words and inferring document distributions in terms of topics. So far, we have recovered the cluster-cluster interaction $A$, which is a noisy expectation of $WW^T$ instead of directly seeking $W$. In our JSMF model, $W$ is unknown and its columns (i.e., example-cluster distributions) are stochastically generated from a known distrubution $f(\alpha)$ goverend by the hyperparameter $\alpha$, rather than sets of parameters to estimate. [1] points out that $W$ is never be able to be recovered in this sense, especially under the limited samples.

Once we recover quality object-cluster matrix B after rectification based on the doubly non-negative geometry of $A_M^*, C^*$, however, we can further try to recover cluster-example matrix $W$ assuming

our recovered $A$ is quite close to $A_M^*$. Since $\mathbb{E}[H_m] = n_m B W_m$ for each example $m$, we can compute $W_m$ by solving the following simplex-constrained Non-Negative Least Square (NNLS) problem:

$$\min_{W_m \in \Delta^K} \|B W_m - H_m/n_m\|$$

This optimization can be solved via exponentiated gradient algorithm similar to what we use for recovering object-cluster matrix $A$. Analogously, it can be easily parallelized by per-document fashion because we are soloving independent optimization for each document given inferred $B$.

## 4  Experimental Results

**Qualitative results.** The 15 clusters from the Movies dataset is attached at the end. One can verify that while Gibbs learns slightly better clusters, AP's results are comparable, whereas Baseline algorithm learns nothing.

**Quantitative results.** The following shows full results from real experiments.

**Legality** ($\sum_{k=1}^K \sum_{l=1}^K A_{kl} = \sum_{k=1}^K \sum_{l=1}^K p(Z_1 = k, Z_2 = l)$) assesses how close the recovered cluster-cluster matrix $A$ is to a legal joint distribution whose entries sum to 1. The results show that the recovered $A$ becomes close to a legal joint distribution under the DC and AP rectification, whereas the entry sum for Baseline is far higher than 1. Note that we intentionally avoid projecting the recovered $A$ down to $\mathcal{DNN}_K$ in order to verify the quality of our new recovery algorithm in terms of legality. **Validity** ($KL_{sym}\left(\sum_{k=1}^K p(Z_1, Z_2 = k) \| \sum_{i=1}^N p(Z_1|W_1 = i)p(W_1 = i)\right)$) gauges the discrepancy between two different constructions of the marginal $p(Z)$: column-sum of $A$ vs applying Bayes' rule. As shown in the results, DC and AP eliminate the discrepancies between two different constructions. Note that the behavior is similar to Legality because marginal construction from an illegal $A$ could be a source of discrepancies.

**Synthetic experiments.** With the same vocabulary curation, we generate (10,000, 25,000, 50,000) semi-synthetic copora from the models trained with 50/150/100/100 synthetic anchors for NIPS, NYTimes, Movies and Songs, respectively. We sampled documents with 300 tokens for each dataset

from a Dirichlet with symmetric hyperparameters 0.03. The following shows full results on semi-synthetic data with $M = 50,000$ corresponding to real experiments. You can verify AP and Gibbs are comparable, but the gaps against the Baseline algorithm are lower than what we have seen in real experiments. This is because semi-synthetic data is generated from the model, whereas the real data in practice never precisely follows the model.

We also measure several parametric gaps between the learned matrices and the truth matrices that we used for generating semi-synthetic documents varying two different sizes of documents.

We can verify that AP not only learns better topics $B$ and their interactions $A$ than the Baseline algorithm, but also increases the anchor recovery rates. In addition, as we have more documents, the gaps between AP and the Baseline algorithm decrease. We are also showing how well our cluster-example recovery proposal works, and how much the result is consistent to the learned cluster interaction by the panels in the second column. (Difference measure is Frobenius norm, but symmetric KL-divergence shows the same behaviors.)

## 5  Analysis of Algorithm

**AP Convergence.**   Figure 1 shows the actual convergence behavior on NYTimes. For 100 iterations, red solid line and blue dashed line illustrate $\log_{10}(\|C' - C\|_F)$ and the logarithm of the average distance to each of three sets, respectively.[1]

Figure 1: Locally linear convergence of AP.

**How does DC work?**   Diagonal completion is a special case of matrix completion, which is often solved by minimizing the nuclear norm consistent over matrices with the known data. While first-order methods like [7] generally provide an effective solution to general matrix completion, our alternative algorithm takes advantage of the specific structure of the diagonal completion problem. Suppose we bisects vocabulary into $\mathcal{V}_1$ and $\mathcal{V}_2$. Then it yields a block factorization.

$$\begin{bmatrix} C_{11} & C_{12} \\ C_{21} & C_{22} \end{bmatrix} = \begin{bmatrix} B_1 \\ B_2 \end{bmatrix} A \begin{bmatrix} B_1^T & B_2^T \end{bmatrix} = \begin{bmatrix} B_1 A B_1^T & B_1 A B_2^T \\ B_2 A B_1^T & B_2 A B_2^T \end{bmatrix}$$

Assume that each cluster associates with a sufficient number of objects, being distinguishable only with either $\mathcal{V}_1$ or $\mathcal{V}_2$. It means neither $B_1$ nor $B_2$ should be (nearly) rank deficient. Under this assumption, we find a basis for the range space of $C_{12}$ from the leading singular vectors, then fill the diagonal elements of $C_{11}$ to minimize the distance from $B$ to this subspace. However, in practice, the co-occurrence matrix is contaminated: rather having $C_{12} = U_1 \Sigma_{12} V_2^T$, we actually have $\hat{C}_{12} = C_{12} + E_{12} = \hat{U}_1 \hat{\Sigma}_{12} \hat{V}_2^T$.

**Error analysis for DC.**   In practice, the co-occurrence matrix is contaminated: rather having $C_{12} = U_1 \Sigma_{12} V_2^T$, we actually have $\hat{C}_{12} = C_{12} + E_{12} = \hat{U}_1 \hat{\Sigma}_{12} \hat{V}_2^T$.

Using Wedin's second $\sin\Theta$ theorem (Stewart, V.4.1, Th 4.4) and norm bounds, we have that the maximum sine between the desired space $U_1$ and the computed space $\hat{U}_1$ is $\|\sin\Theta\|_2 \leq \|E_{12}\|/\hat{\sigma}_k \equiv \gamma$ where $\hat{\sigma}_k$ is the smallest retained singular value of the empirical block $\hat{C}_{12}$ and $\|E_{12}\|$ is the magnitude of the difference. This leads to the bounds

$$\min_{W^T W = I} \|U_1 W - \hat{U}_1\| \leq \sqrt{\frac{2\|E_{12}\|}{\hat{\sigma}_k}} \equiv \sqrt{2\gamma}, \tag{9}$$

$$\min_{W^T W = I} |e_j^T U_1 W - e_j^T \hat{U}_1| \leq \gamma. \tag{10}$$

Based on he diagonal reconstruction formula (Equation (11) in the main paper), our noisy diagonal will be $\hat{d}_j = \hat{u}_j^T \hat{L}_{*j}/(1 - \hat{u}_j^T \hat{u}_j)$. We write the ratio between the approximate and true values as

$$\frac{\hat{d}_j}{d_j} = \left( \frac{\hat{u}_j^T \hat{L}_{*j}}{u_j^T L_{*j}} \right) \left( \frac{1 - u_j^T u_j}{1 - \hat{u}_j^T \hat{u}_j} \right). \tag{11}$$

The former term in the product can be written $1 + \delta_1$ with

$$|\delta_1| \lesssim \frac{\|L_{*j} - \hat{L}_{*j}\| + \|E_{11}\|\|L_{*j}\|}{u_j^T L_{*j}}$$

and the latter term can be bounded as $1 + \delta_2$ with $|\delta_2| \leq \gamma/(1 - \|\hat{u}_j\|^2)$.

# 6   Related and Future Work

Through this paper, we examine why rectification is necessary, proposing two novel rectification algorithms. Whereas AP enfoces every desirable property, Diagonal Completion (DC) enforces only low-rank property on top of joint-stochasticty without requiring positive semi-definiteness. While the results show AP is the appropriate method for our configuration, DC can be also useful for other tasks based on the co-occurrence matrix. For example, many different embeddings based on the co-occurrence statistics have their own treatment to the diagonal entries, but most of them are based on simple heuristics or trial-and-error approahces rather than strictly enforcing certain mathematical structures. Therefore one might test our DC toward their co-occurrence statistics if low-rank structure is suitable for their tasks.

On the other side, AP finds better anchors as well as performs better inference for learning topics and their interactions. We conjecture that AP's treatment on bursty and rare words smoothens noisy eccentric vertices on the co-occurrence space $\overline{C}$, making most objects to be well spread out inside the convex rather than being crowded. (The first figure on the main paper shows that rectified space is significantly smaller than the original space in terms of the area, but object vertices in general well spread out through the space, giving better and clear cue of a convex shape.) Therefore, extereme vertices in this smooth space are likely to be truly informative, well summarizing other objects based on the underlying topic interactions.

## Footnotes

[1]While AP performs an alternating projection, we also keep track of the average of the projected points and the distance to each set per iteration for validation purpose.

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

| Arora et al. 2013 (Baseline) | This paper (AP) | Probabilistic LDA (Gibbs) |
|---|---|---|
| Pulp Fiction (1994) | Aladdin (1992) | Beauty and the Beast (1991) |
| Silence of the Lambs (1991) | Toy Story (1995) | Aladdin (1992) |
| Shawshank Redemption (1994) | Beauty and the Beast (1991) | Mary Poppins (1964) |
| Forrest Gump (1994) | Babe (1995) | Lion King (1994) |
| The Fugitive (1993) | Lion King (1994) | Little Mermaid (1989) |
| Pulp Fiction (1994) | Shrek (2001) | Lord of the Rings I (2001) |
| Forrest Gump (1994) | Lord of the Rings II (2002) | Lord of the Rings II (2002) |
| Silence of the Lambs (1991) | Austin Powers (1990) | Matrix (1999) |
| Shawshank Redemption (1994) | Lord of the Rings I (2001) | Lord of the Rings III (2003) |
| The Fugitive (1993) | Lord of the Rings III (2003) | Shrek (2001) |
| Pulp Fiction (1994) | Star Wars V (1980) | Alien (1979) |
| Silence of the Lambs (1991) | Star Wars IV (1977) | Aliens (1986) |
| Shawshank Redemption (1994) | StarWars IV (1983) | Blade Runner (1982) |
| Forrest Gump (1994) | Indiana Jones (1981) | Army of Darkness (1993) |
| The Fugitive (1993) | Terminator (1984) | Star Trek II (1982) |
| Pulp Fiction (1994) | Independence Day (1996) | Independence Day (1996) |
| Shawshank Redemption (1994) | Twister (1996) | The Rock (1996) |
| Silence of the Lambs (1991) | The Rock (1996) | Mission Impossible (1996) |
| Forrest Gump (1994) | Mission (1996) | Twister (1996) |
| Braveheart (1995) | Broken Arrow (1996) | Toy Story (1995) |
| Pulp Fiction (1994) | Apollo 13 (1995) | Apollo 13 (1995) |
| Shawshank Redemption (1994) | Dances with Wolves (1990) | The Fugitive (1993) |
| Silence of the Lambs (1991) | True Lies (1994) | Dances with Wolves (1990) |
| Forrest Gump (1994) | Pulp Fiction (1994) | Forrest Gump (1994) |
| The Fugitive (1993) | Batman (1989) | Pulp Fiction (1994) |
| Pulp Fiction (1994) | Shawshank Redemption (1994) | Maltese Falcon (1941) |
| Silence of the Lambs (1991) | Matrix (1999) | African Queen (1951) |
| Shawshank Redemption (1994) | Silence of the Lambs (1991) | Key Largo (1948) |
| Forrest Gump (1994) | Pulp Fiction (1994) | Double Indemnity (1944) |
| Star Wars Episode IV (1977) | Lord of the Rings I (2001) | American Graffiti (1973) |
| Pulp Fiction (1994) | Pulp Fiction (1994) | Remains of the Day (1993) |
| Silence of the Lambs (1991) | Shawshank Redemption (1994) | Much Ado about Nothing (1993) |
| Shawshank Redemption (1994) | Usual Suspect (1995) | Copycat (1995) |
| Forrest Gump (1994) | The Piano (1993) | The Piano (1993) |
| The Fugitive (1993) | Sense and Sensibility (1995) | What's Eating Gilbert Grape (1993) |
| Pulp Fiction (1994) | American Beauty (1999) | American Beauty (1999) |
| Silence of the Lambs (1991) | Sixth Sense (1999) | Sixth Sense (1999) |
| Shawshank Redemption (1994) | Austin Powers (1999) | Gladiator (2000) |
| Forrest Gump (1994) | American Pie (1999) | American Pie (1999) |
| The Fugitive (1993) | Shakespeare in Love (1998) | Fight Club (1999) |
| Pulp Fiction (1994) | Forrest Gump (1994) | Amelie (2001) |
| Silence of the Lambs (1991) | Jurassic Park (1983) | Mulholland Drive (2001) |
| Shawshank Redemption (1994) | Mrs. Doubtfire (1993) | Lost in Translation (2003) |
| Forrest Gump (1994) | Pretty Woman (1990) | Adaptation (2003) |
| The Fugitive (1993) | Ghost (1990) | Memento (2000) |
| Pulp Fiction (1994) | Godfather (1972) | Godfather (1972) |
| Silence of the Lambs (1991) | One Flew Over the Cuckoo's Nest (1975) | Indiana Jones (1981) |
| Shawshank Redemption (1994) | Casablanca (1942) | Casablanca (1942) |
| Forrest Gump (1994) | Godfather II (1974) | One Flew Over the Cuckoo's Nest (1975) |
| The Fugitive (1993) | Annie Hall (1977) | Star Wars V (1980) |
| Pulp Fiction (1994) | Titanic (1997) | Hunt for Red October (1990) |
| Silence of the Lambs (1991) | The Game (1997) | The Rock (1996) |
| Shawshank Redemption (1994) | Liar, Liar (1997) | Die Hard 2 (1990) |
| Forrest Gump (1994) | Chasing Amy (1997) | Face Off (1997) |
| The Fugitive (1993) | Scream (1996) | Air Force One (1997) |
| Pulp Fiction (1994) | Tombstone (1993) | Ferris Bueller's Day Off (1986) |
| Silence of the Lambs (1991) | The Specialist (1994) | Breafast Club (1985) |
| Shawshank Redemption (1994) | Judge Dredd (1995) | Airplane (1980) |
| Forrest Gump (1994) | Leon (1994) | Big (1988) |
| The Fugitive (1993) | Species (1995) | Christmas Story (1983) |

| Arora et al. 2013 (Baseline) | This paper (AP) | Probabilistic LDA (Gibbs) |
|---|---|---|
| Pulp Fiction (1994) | Pulp Fiction (1994) | Tee Departed (2006) |
| Silence of the Lambs (1991) | Silence of the Lambs (1991) | Casino Royale (2006) |
| Shawshank Redemption (1994) | Usual Suspects (1995) | Little Miss Sunshine (2006) |
| Forrest Gump (1994) | 12 Monkeys (1995) | V for Vendetta (2006) |
| The Fugitive (1993) | Seven (1995) | Batman Begins (2005) |
| Pulp Fiction (1994) | Star Wars IV (1977) | Spider-Man (2002) |
| Silence of the Lambs (1991) | Star Wars Episode IV (1983) | Ocean's Eleven (2001) |
| Shawshank Redemption (1994) | Jerry Maguire (1996) | Harry Potter I (2001) |
| Forrest Gump (1994) | Godfather (1972) | Lord Of the Rings I (2001) |
| The Fugitive (1993) | Time to Kill (1996) | My Big Fat Greek Wedding (2002) |
| Pulp Fiction (1994) | Fargo (1996) | Fargo (1996) |
| Silence of the Lambs (1991) | Leaving Las Vegas (1995) | Shakespeare in Love (1998) |
| Shawshank Redemption (1994) | Dead Man Walking (1995) | Good Will Hunting (1997) |
| Forrest Gump (1994) | The Postman (1994) | L. A. Confidential (1997) |
| The Fugitive (1993) | Trainspotting (1996) | Full Monty (1997) |