[Reviews · NeurIPS 2015]

Submitted by Assigned_Reviewer_1

Summary of the paper: This paper improves Anchor Word topic models [1,2] with rectifying operations in order to satisfy a newly introduced double non-negativity constraint, which is proved to be necessary by authors. Experimental results present the rectification indeed improves various aspects of topic modeling performance.

Quality: Although most contents of the paper feel solid, experimental part can be further improved. Figure 4, in particular, have a lot of missing data points and too small to read it easily. I also want to encourage authors to provide training time comparisons because one of the largest benefit of using [1,2] was computation even though I feel the rectification will not require extreme computation.

Clarity: Mostly clear.

Originality: I believe authors are the first to employ rectification in Anchor Word topic models. Even their method appears to be a straight-forward extension, authors provided enough discussions and theories that justify their idea in a broad spectrum.

Significance: Although their idea will not begin an entirely new topic modeling methodology, I feel their algorithm can be easily employed various NMF models and can be directly utilized in large scale topic modeling as authors suggested in Section 6.

[1] Arora, S., Ge, R., & Moitra, A. (2012, October). Learning topic models--going beyond SVD. In Foundations of Computer Science (FOCS), 2012 IEEE 53rd Annual Symposium on (pp. 1-10). IEEE.

[2] Arora, S., Ge, R., Halpern, Y., Mimno, D., Moitra, A., Sontag, D., & Zhu, M. (2012). A practical algorithm for topic modeling with provable guarantees. arXiv preprint arXiv:1212.4777.
Summary: The paper founds a theoretical issue in Anchor Word topic models [1,2] and suggests a solution. Even though the paper can be improved further on a few aspects, it is still solid and will have a noticeable impact in machine learning communities. (see below for references)

Submitted by Assigned_Reviewer_2

The paper focuses on improving the performance of spectral inference for topic models. The paper describes the problem with existing spectral inference methods using the framework of Joint Stochastic Matrix Factorization (JSMF) and then proposes a method to rectify it. Experimental results show the proposed method achieve good results on four datasets.

The paper tackles an important problem of improving existing spectral learning methods for topic models. I think the paper is well written, the technique is sound and might inspire additional work on this direction. Here are my additional comments:

- It would make the paper much more approachable if it can explicitly list the conditions under which existing techniques (e.g., Arora et al, ICML'13) do not perform well and the method proposed in this paper is needed.

- I am wondering why more commonly used evaluation metrics like topic coherence are not used for comparing performances.
Summary: The paper addresses an important problem of improving the robustness of spectral methods for topic models. I enjoy reading the paper and think that it might inspire additional work along this direction.

Author Feedback
Author rebuttal: We thank the reviewers for their valuable comments and suggestions for further improvement.

Training time: AP rectification is inexpensive due to the truncated eigen-decomposition. While we used 150 iterations of Alternating Projection (AP) in the experiments, the co-occurrence matrix usually converges more quickly (within 30 iterations).

When does the previous algorithm fail? The Arora et al's algorithm fails in three cases (a) when word co-occurrence distributions are noisy or poorly estimated due to the limited number of documents, which is typical for real data, (b) when there is any degree of interaction between topics, the previous algorithm is unable to incorporate it into the inference, because the anchors were inevitably selected as too rare words to co-occur with other anchors -- manual frequency cutoffs are necessary for even moderately acceptable performance --, and (c) when the number of topics is not sufficiently large, an early choice of poor anchors impedes better choices in the future. We will further clarify these three points to clearly articulate the contribution of this paper.

Topic Coherence: This metric is shown in the rightmost column of the Figure 4 simply as a "Coherence".

Small datasets: We have verified that we get similar results on both smaller corpora and smaller sizes of vocabularies.

We are excited about our work because:

(1) The previous work (Arora et al) does not capture topic correlations (even for large numbers of topics), losing the latent interactions between topics. Our rectification fixes this problem.

(2) The previous work fails to learn quality topics unless the number of topics is large enough. Our rectification resolves this issue, making spectral inference comparable to probabilistic LDA.